# Impact of Feeding Probiotics on Blood Parameters, Tail Fat Metabolites, and Volatile Flavor Components of Sunit Sheep

**DOI:** 10.3390/foods11172644

**Published:** 2022-08-31

**Authors:** Ting Liu, Taiwu Zhang, Yanni Zhang, Le Yang, Yan Duan, Lin Su, Jianjun Tian, Lina Sun, Bohui Wang, Ye Jin

**Affiliations:** 1College of Food Science and Engineering, Inner Mongolia Agricultural University, Hohhot 010018, China; 2Ordos City Inspection and Testing Center, Ordos 017000, China

**Keywords:** probiotics, blood indicators, tail fat metabolites, volatile flavor compounds

## Abstract

Sheep crude tail fat has unique nutritional values and is used as a raw material for high-quality natural oil. The purpose of this study was to investigate the effects of probiotics on the metabolites and flavor of sheep crude tail fat. In this study, 12 Sunit sheep were randomly divided into an experimental group (LTF, basal feed + *Lactiplantibacillus*
*plantarum* powder) and a control group (CTF, basal feed). The results of sheep crude tail fat analysis showed that blood lipid parameters were significantly lower and the expression of fatty acid synthase and stearoyl-CoA desaturase genes higher in the LTF group than in the CTF group (*p* < 0.05). Metabolomic analysis via liquid chromatography–mass spectrometry showed that the contents of metabolites such as eicosapentaenoic acid, 16-hydroxypalmitic acid, and L-citrulline were higher in the LTF group (*p* < 0.01). Gas chromatography–mass spectrometry detection of volatile flavor compounds in the tail fat showed that nonanal, decanal, and 1-hexanol were more abundant in the LTF group (*p* < 0.05). Therefore, *Lactiplantibacillus plantarum* feeding affected blood lipid parameters, expression of lipid metabolism-related genes, tail fat metabolites, and volatile flavor compounds in Sunit sheep. In this study, probiotics feeding was demonstrated to support high-value sheep crude tail fat production.

## 1. Introduction

Sunit sheep are a breed of Mongolian sheep with excellent meat quality that live in the grasslands of Inner Mongolia [1]. Local shepherds have traditionally eaten sheep crude tail fat. The fat content of the sheep’s tail can reach 5–15% of the sheep’s slaughter weight [2]. Sheep crude tail fat, a byproduct of lamb processing, is often used industrially in the manufacturing of lubricants and surfactants. Sheep crude tail fat is the subject of few reports by food flavor researchers. Most previous experiments have investigated the gene that affects sheep tail type, while few studies have explored its usage [3]. Additionally, some previous research focused on the composition of sheep tail fat, causes of tail fat deposition, and differences in tail fat among breeds or between sexes [4,5]. In contrast, tail fat metabolism and flavor differences associated with the use of supplements in the same basal diet have not been investigated thoroughly.

Metabolomics can be used to analyze heterologous metabolites between an experimental group and a control group to screen for characteristic markers, key enzymes, and genes associated with the experimental manipulation. Probiotics are increasingly used in animal feed as a feed supplement. Research on probiotic supplementation in ruminants is scarce [6]. Previous studies have shown that probiotics are living microorganisms that regulate the host’s metabolism via modulation of the composition and function of the intestinal microbiota. *Lactiplantibacillus plantarum* is a probiotic that affects host metabolism. For example, intake of *Lactiplantibacillus plantarum* in a hyperlipidemic rat model effectively reduced plasma total cholesterol (TC), triglyceride (TG), and low-density lipoprotein (LDL) levels [7]. In addition, *Lactiplantibacillus plantarum* may regulate lipogenesis and catabolism-related genes to reduce the level of fat accumulation in the liver of rats fed a high-fat diet [8]. Therefore, *Lactiplantibacillus plantarum* is a potential therapeutic agent against hyperlipidemia in both animals and humans [9].

Fatty flavor is an important sensory indicator affecting the perception of food quality. Meat flavors differ among the different body parts of livestock and poultry [10]. Here, we used headspace solid-phase microextraction combined with gas chromatography–mass spectrometry to clarify the effects of *Lactiplantibacillus plantarum* supplementation on the flavor of Sunit sheep fat. Sheep crude tail fat has research value, as it is a table food. Probiotics may affect animal sensory and nutritional indicators via regulation of blood lipid indicators and gene expression (Figure 1). In this experiment, Sunit sheep were used as the study subject to assess the effect of *Lactiplantibacillus plantarum* on tail fat metabolism to increase the value of sheep byproducts used in the food industry.

## 2. Materials and Methods

### 2.1. Animal Experiments and Feeding Methods

Animal experiments were conducted by the guidelines for animal experiments of the National Institute of Animal Health of China (GB 14925-2010) and were approved by the Ethics Committee of Inner Mongolia Agricultural University (NND2021072). Twelve 3-month-old healthy and purebred Sunit sheep were randomly selected from the Chuanjing Sumu Haratu Gacha area in the Urad Middle Banner of Bayannaoer City, Inner Mongolia. They were randomly divided into the control group (CTF) and the *Lactiplantibacillus plantarum* group (LTF). There were 6 Sunit sheep in each group, half of them male and half female, with average live weights of 20.73 ± 0.76 kg (CTF) and 21.27 ± 1.41 kg (LTF), respectively. There was no statistical difference in live weight between the two feeding groups (*p* = 0.746). CTF was fed the basal diet, and LTF was fed the basal diet supplemented with *Lactiplantibacillus plantarum* powder 12 g/d (Shandong Baolai Lilai Bioengineering Co., Ltd.). The viable count of *Lactiplantibacillus plantarum* was 3 × 10^10^ cfu/g. 12 g of bacterial powder, which was mixed with 60 mL of water in advance, and 10 mL was drawn with a syringe to feed each sheep (fed Lactiplantibacillus plantarum once a day). They were allowed to drink water and move freely without any antibiotics during the animal experiment.

The basal diet composition and nutrient level of experimental sheep were as follows: Basic diet twice a day per sheep. Raw material composition: silage corn (60.0%), sunflower meal (18.0%), soybean cake (13.5%), cottonseed meal (4.0%), ground limestone (1.0%), sodium chloride (0.8%), calcium hydrophosphate (0.5%), Lys (0.8%), vitamin and mineral premixes (1.5%, Mineral vitamin premixes were provided with 4000 IU of vitamin A, 3500 IU of vitamin K3, 2500 IU of vitamin B12, 1500 IU of vitamin D3, and 35 mg of vitamin E per kg of diet; Zinc 35 mg, iron 35 mg, manganese 20 mg, copper 12 mg, iodine 1.4 mg, cobalt 0.2 mg). Nutrient level: crude protein (15.9%), crude fat (5.6%), neutral detergent fiber (26.8%), acid detergent fiber (17.6%), crude ash (7.8%), calcium (1.2%), phosphorus (0.6%), lysine (0.5%), metabolizable energy (11.57 MJ/kg).

### 2.2. Sample Collection at Slaughter

After the 90-day feeding experiment, the Sunit sheep were sent to Grassland Hengtong Food Co., Ltd., which is 38 km away from the feeding place. Sunit sheep were held off feed and rest management for 12 h before slaughter. Blood samples were collected from 12 Sunit sheep in the morning before feeding, and sheep jugular vein blood samples were collected and stored in EDTA tubes at −4 °C. Blood samples were centrifuged at 3000× *g* for 10 min at 4 °C in the laboratory, and plasma was stored at −80 °C until testing. About 50 g of tail fat was collected immediately after fur removal. On the one hand, the tail fat was cut into 0.5 cm × 0.5 cm × 1 cm fat pieces and placed in a sterile 2 mL gene tube without enzymes for RNA analysis and metabolite analysis. The tail fat gene samples were frozen in liquid nitrogen and stored at −80 °C until analysis. On the other hand, the tail fat was placed at −20 °C to detect the volatile flavor compounds of sheep crude tail fat.

### 2.3. Determination of Blood Lipid Parameters in Sunit Sheep

Triglyceride (TG), total cholesterol (TC), high-density lipoprotein (HDL), and low-density lipoprotein (LDL) in blood were measured by spectrophotometry according to the instruction manual of the assay kit provided by the Nanjing Jiancheng.

### 2.4. Lipid Metabolism Genes in the Sunit Sheep Crude Tail Fat

#### 2.4.1. RNA Extraction and RT-PCR

Sample total RNA was extracted from −80 °C frozen tissues using RNAiso Plus (Takara, Dalian) following the manufacturer’s instructions. The concentration and purity of total RNA in the sample were checked using BioDrop μLite (BioDrop, Cambridge, England), and RNA integrity was verified by agarose gel electrophoresis. Genomic DNA was removed with gDNA Eraser (Takara, Dalian). cDNA was synthesized using M-MLV Reverse Transcriptase (Takara, Dalian).

#### 2.4.2. Quantitative PCR

The SYBR Premix Ex Taq dye real-time quantitative PCR kit (Takara, Dalian) was used in an ABI 7500 real-time PCR instrument (Applied Biosystems, Waltham, MA, USA). β-actin was used as the internal reference gene, and the primer sequences are shown in Appendix A. The reaction (25 μL) consisted of a 2 μL mixed cDNA template (50 ng/μL), 12.5 μL SYBR Premix Ex Taq (Takara), 2 μL forward and reverse primers (5 μM), and 8.5 μL DNase/RNase free water. qPCR conditions: 38 cycles of 50 °C/2 min, 95 °C/10 min, 95 °C/15 s, and 60 °C/1 min, followed by amplicon dissociation (95 °C/15 s, 60 °C/15 s, 95 °C/15 s). Reference Livak was used to calculate mRNA expression using the comparative CT method (2^−^^ΔΔCt^) [11].

### 2.5. Determination of Sunit Sheep Crude Tail Fat Metabolites

To obtain the metabolites from tail fat samples, 1 mL of pre-chilled extraction solvent methanol/acetonitrile/H_2_O (2:2:1, *v*/*v*/*v*) was added to an 80 mg sample and adequately vortexed. The lysate was homogenized by MP homogenizer (24 × 2, 6.0 M/S, 60 s, twice) and sonicated at 4 °C (30 min/once, twice), then centrifuged at 14,000× *g* for 20 min at 4 °C, and the supernatant was dried in a vacuum centrifuge at 4 °C. For LC-MS evaluation, the samples were re-dissolved in 100 μL acetonitrile/water (1:1, *v*/*v*) solvent.

#### 2.5.1. LC-MS Analysis of Tail Fat Metabolites

For untargeted metabolomics of polar metabolites, the solution was investigated using a quadrupole time-of-flight mass spectrometer (Sciex TripleTOF 6600). The aid of electrospray ionization is essential to combine the solution under study with HIC (hydrophilic interaction chromatography) in this device. The chromatographic column used for the separation of the liquid phase was the ACQUITY UPLC BEH Amide column. The dimensions of this column were 2.1 mm × 100 mm, with a particle size of 1.7 µm (waters, Ireland). Three solvents were used in the separation: solvent A, 25 mM ammonium acetate and 25 mM ammonium hydroxide in water, and solvent B, acetonitrile. These two types of three solvents were used as gradient solvents to participate in the separation process of the liquid phase. Solution B changed as follows during the separation process: first with a concentration gradient of 85% for 60 s; second, over 11 min, maintaining a linear change to 65%; then, decreasing more rapidly over 6 s to 40% and held at this concentration for 4 min; finally, the concentration was increased to 85% in 6 s at high speed. At this point, we entered a 5 min re-equilibration period.

During the injection process, the flow velocity of the sample was kept at 0.4 mL per minute, the column temperature was 25 degrees Celsius, and the temperature of the automatic sampling device was 5 degrees Celsius. The total injection volume of the whole injection process was 2 μL. The mass spectrometer operates in two modes, namely, negative ionization and positive ionization modes. This is to separate and identify substances with different points after ionization. The conditions of the ESI source were set according to these items: ion source gas 1 was gas 1, set to 60; ion source gas 2 was gas 2, also set to 60; for Curtain gas (CUR), set to 30; source temperature set at 600 degrees Celsius under conditions; IonSpray Voltage Floating (ISVF) range was ±5500 volts. In the acquisition of results by MS (mass spectrometer), the acquisition range of the results was set in the m/z range of 60–1000 Da in this experiment. For the scan time of TOF MS, the accumulation was 0.20 s per spectrum. In the automatic MS/MS result acquisition, this experiment was carried out with the result acquisition range of the device set in the m/z range of 25–1000 Da. For this acquisition process, the cumulative time for the production scan was set to 0.05 s per spectrum. The high-sensitivity mode of information-dependent acquisition (IDA) was used to acquire the results of production scans. According to the purpose and equipment of this experiment, some constraints were set: the fixed collision energy was 35 V, and the fluctuation did not exceed 15 eV; the declustering potential (DP) was 60 V (+) and −60 V (−); for the isotope exclusion conditions, set to exclude substances within 4 Da; to select candidate ions, the number of candidate ions to be detected in each cycle was 10. The LC-MS experiments were commissioned by Shanghai Applied Protein Technology Co., Ltd.

#### 2.5.2. Tail Fat Metabolite Analysis

For the raw data obtained by MS, because the format was a wiff scan file, to import it into XCMS, an indispensable pre-step was to convert it in ProteoWizard MSConvert, and the wiff file became an MzXML file. During peak picking, some necessary conditions needed to be set, including: centWave m/z = 25 ppm, peak width = c (10, 60), prefilter = c (10, 100). In addition, for peak grouping, filter conditions were: bw = 5, mzwid = 0.025, minfrac = 0.5. In the experiment, the necessary condition for ion feature extraction was that more than half of the obtained data in at least one group were non-zero before they could be used. The compound composition of the identified metabolites was classified and identified by MS/MS spectra. The built-in database used in this process also had its standards for building and was also compared with the NIST database. The data obtained in the above steps were normalized to convert the data into the intensity of the total peak. The data were then subjected to multivariate analysis including principal component analysis (PCA) and orthogonal partial least-squares discriminant analysis (OPLS-DA). These analyses were performed in SIMCA-P (version 14.1, Umetrics, Umea, Sweden) software. To evaluate the robustness of the model, it was essential to use 7-fold cross-validation and response permutation testing. The VIP value (variable importance in the projection) of each variable in the OPLS-DA model was used to assess the contribution of each variable to the overall classification. Statistical criteria are as follows: the experiment used unpaired Student’s *t*-test. The results obtained have a *p*-value < 0.05 and a VIP-value > 1, which were considered statistically significant. A total of 24 metabolites were detected after excluding duplicates under the screening of positive and negative ion models.

#### 2.5.3. Functional Predictive Analysis of Tail Fat Metabolites

For KEGG (Kyoto Encyclopedia of Genes and Genomes) pathway annotation, the metabolites were blasted against the Kyoto Encyclopedia of Genes and Genomes (KEGG) database to fetch back their COs and were afterward mapped to pathways in KEGG11. The correspondent KEGG pathways were obtained. To further research the influence of differentially expressed metabolites, enrichment analysis was conducted. KEGG pathway enrichment analyses were exercised based on Fisher’s exact test, considering the whole metabolites of each pathway as a background dataset. In addition, only pathways with *p*-values under a threshold of 0.05 were considered as significantly changed pathways. For hierarchical clustering, Cluster 3.0 (13 October 2021, http://bonsai.hgc.jp/~mdehoon/software/cluster/software.htm) and the Java Treeview software (13 October 2021, http://jtreeview.sourceforge.net) were utilized. Euclidean distance was used in the present study to calculate similarity measures and the average linkage clustering (clustering used the centroids of the observations), for clustering was chosen when performing hierarchical clustering. The heatmap is often presented as a visual aid.

### 2.6. Determination of Volatile Flavor Compounds in Tail Fat

Volatile compounds were analyzed through solid-phase microextraction (SPME) and GC/MS via placing 5 g of minced Sunit tail fat in a 15 mL PTFE (polytetrafluoroethylene) septum-sealed headspace sample vial. Headspace sampling was performed after the vial was extracted at 50 °C with shaking for 45 min. The previously exposed SPME fibers were heated in the headspace of the headspace vial at 250 °C for 30 min at the GC inlet. Volatiles were thermally desorbed to the front of the capillary column (TR-5MS 30 m × 0.25 mm; film thickness 0.25 μm; Thermo Fisher Scientific).

The carrier gas used in the instrument was helium at a flow rate of 1.0 mL/min, and the injector was splitless. The heating method used was as follows: first, 40 °C hold 3 min; second, ramp to 150 °C at 4 °C per minute and hold 1 min; third, 5 °C/min to 200 °C; finally, 10 °C/min to 250 °C, subsequently hold 5 min. The transfer line and ion source temperature were kept at 250 degrees Celsius. MS was acquired at 70 eV, and the scan range was 30 to 400 m/z. After everything was ready, the pretreated sample was injected into the gas chromatography-mass spectrometer (GC-MS) system for 4 min of desorption.

The final analysis was carried out with the aid of the Trace 1300 series GC gas chromatograph that came with an ISQ mass spectrometer and Xcalibur ChemStation (Thermo Fisher Scientific, Shanghai China), with three replicates (n = 3) for each sample, as described by Luo et al. [12]. Volatile compounds were identified against mass spectral data in the library (NIST MS Search 2.0). The number of compounds was computed by comparing the peak area with that of the internal standard (2-methyl-3-heptanone, obtained from a total ion chromatogram with a response factor of 1).

### 2.7. Statistical Analysis of Experimental Data

Data were analyzed using the SPSS, version 26.0, using analysis of variance (ANOVA) to compare means by the least significant difference (LSD) procedure. Tabular data are expressed as mean and relative standard deviation (RSD). *p* values less than 0.05 were considered statistically significant. Origin version 2021 developed by OriginLab was used for mapping.

## 3. Results

### 3.1. Effects of Probiotics on Plasma Lipid Profiles

The effects of *Lactiplantibacillus plantarum* supplementation on blood lipids in Sunit sheep were analyzed (Figure 2). The concentrations of all four plasma lipid parameters tested were reduced in the sheep supplemented with *Lactiplantibacillus plantarum* powder (LTF group). Of the factors tested, only the TC level did not differ significantly between the two feeding methods (*p* = 0.81). Changes in the ratios of high-density lipoprotein (HDL) to LDL levels and of HDL to TC levels were investigated after supplementation of *Lactiplantibacillus plantarum.* The HDL-to-LDL ratio was significantly higher (Figure 2E) and the HDL-to-TC ratio significantly lower in the LTF group than in the control (CTF) group (Figure 2F) (*p* < 0.05).

### 3.2. Effects of Probiotics on Lipid Metabolism Genes in Tail Fat

Genes related to lipid metabolism, namely, acetyl-CoA carboxylase (ACACA), fatty acid binding protein 4 (FABP4), fatty acid synthase (FASN), fatty acid desaturase 1 (FADS1), fatty acid desaturase 2 (FADS2), peroxisome proliferator activated receptor alpha (PPARA), and stearoyl-CoA desaturase (SCD), were investigated in this study (Figure 3). Among the seven genes assessed in this experiment, the expression levels of FADS1, FASN, and SCD differed significantly between the two groups (*p* < 0.05). The expression of both FASN and SCD was higher while that of FADS1 was lower in the LTF group than in the CTF group.

### 3.3. Correlations between Blood lipid Parameters and Lipid Metabolism Genes

Correlation heatmaps were created to represent interdependencies in the data, indicating correlations between individual metabolites. To identify relationships between blood lipid parameters and lipid metabolism-related genes, correlations were calculated using the Pearson algorithm in the R package “corrplot” (correlation coefficient, r > 0.75). Based on the correlation analysis presented in Figure 4, blood parameters were closely related to lipid synthesis-related regulatory genes. Reduced concentrations of TC, HDL, and LDL in Sunit sheep significantly affected the expression levels of ACACA and FASN and the activity of SCD. These genes are related to catalysis during fatty acid synthesis and regulation of fat deposition around muscles and to the synthesis of palmitate from acetyl-CoA and malonyl-CoA in mammalian cells. Reducing the concentrations of TC, HDL, and LDL significantly inhibited the expression of FABP4 and FADS, impacting the complexing capacity of polyunsaturated fatty acids, as well as the rates of fat and cholesterol deposition. Blood indicators are related to animal health and represent the utilization of animal energy.

### 3.4. Effects of Probiotics on Sheep Crude Tail Fat Metabolites

The instrumental analysis system used for this experiment has great stability, and the experimental data are stable and reliable. After evaluation of six quality control measures, this experiment involved total ion chromatography comparison of the quality control samples, along with principal component analysis. The spectra were overlaid and compared, as shown in Appendix A. The experimental results indicate overlap of the response intensity and retention time of each chromatographic peak, suggesting that the variation caused by instrumental error in the experimental process is small. Metabolites have diverse forms and include many isomers and metabolites with similar molecular masses. Thus, a higher metabolite identification number indicates a lower reliability rating. This project employed the local self-built standard database of Zhongke New Life (Shanghai Applied Protein Technology) as a search library. By comparing the retention time, molecular mass (within <25 ppm error), secondary fragmentation spectrum, collision energy, and other information about metabolites available in the offline database, the structures of metabolites in biological samples could be determined and analyzed. This identification level exceeds Level 2.

Information about all detected differential metabolites is listed in Appendix A. Orthogonal projections to latent structures discriminant analysis (OPLS-DA) is a modification of the partial least-squares discriminant analysis method that filters out noise. OPLS-DA maximizes the difference between groups in t[1], from which variations between groups can be directly distinguished, while variations within groups are reflected on the orthogonal principal component, to[1]. The OPLS-DA model score plot of the two groups shown in Appendix A can be used to distinguish the two groups of samples.

#### 3.4.1. Screening for Differential Metabolites

Variable importance for the projection analysis based on the OPLS-DA model can be employed to measure the impact and explanatory power of the expression patterns of each metabolite on the categorization and discrimination of each group of samples, as well as to mine molecules with biologically meaningful differences. Metabolomics usually uses an OPLS-DA variable importance with a projection value > 1 and *p*-value < 0.05 as strict screening criteria for significantly different metabolites, and these criteria were used in this experiment. A total of 24 significantly different metabolites were identified in this study (Table 1). Compared with the CTF group, 16 metabolites (F5–7, F10–19, F21, F22, and F24) were significantly elevated, and the remaining 8 metabolites (F1–4, F8–9, F20, and F23) were significantly reduced in the LTF group. The identified metabolites are mainly involved in the metabolism of lipids, nucleotides, and amino acids, with nearly 50% closely related to fat metabolism. Among them, eight metabolites that were extremely different between the two groups are related to lipid metabolism (*p* < 0.01). Organic acids and their derivatives contribute greatly to flavor, and some medium-chain organic acids also affect mitochondrial energy metabolism.

#### 3.4.2. Lipid Metabolite Correlation Analysis

Correlation analysis is useful for elucidating the mutual regulatory relationships among metabolites associated with changes in biological conditions. Correlated metabolites may cooperate in some biological processes (functional correlation). In addition, synergistic and mutually exclusive relationships can exist among metabolites. As an illustration, positively correlated metabolites may indicate metabolites that originate from the same synthetic pathway while negative correlations indicate synthetic transition relationships. As shown in Figure 5, both chord diagrams and network diagrams showed pairs of metabolite molecules with a correlation coefficient |r| > 0.8 and *p* < 0.05. Chord diagrams can show the correlations for numerous metabolites. In positive ion mode, aside from lipid molecules, all other molecular substances showed correlations. In negative ion mode, lipid molecules and organic acids were also related.

In positive ion mode, the lipid metabolite 16-hydroxypalmitic acid (HPA) was significantly and positively correlated with eicosapentaenoic acid (EPA), and both were significantly and negatively correlated with hepatic phosphatidylcholine (PC). The omega-3 fatty acid EPA has been found to reduce oxidized LDL levels in patients with hypertriglyceridemia, contribute to membrane stability, improve lipoprotein clearance, and reduce inflammation [13]. Increasing the content of the lipid metabolite HPA or reducing the PC content leads to increased EPA.

In negative ion mode, organic acids such as succinate were positively correlated with ketoisocaproic acid, while ketoisocaproic acid was negatively correlated with the lipids deoxycholic acid (DCA) and chenodeoxycholate. Succinate activates uncoupling protein 1 in brown adipose tissue in vivo, causing mitochondrial oxidative respiratory electron transfer and breakdown of adenosine triphosphate to generate heat, thereby enhancing glucose tolerance. Elevated contents of branched-chain amino acids and their metabolites are related to insulin resistance. Previous studies have demonstrated that the leucine metabolite alpha ketoisocaproate inhibits insulin-stimulated glucose transport in myotubes. Interventions that increase branched-chain amino acid catabolism promote muscle glucose utilization and improve insulin resistance [14]. DCA is positively correlated with chenodeoxycholate, and they act synergistically in physiological processes.

#### 3.4.3. KEGG Annotation and Metabolic Pathway Analysis of Differential Metabolites

The Kyoto Encyclopedia of Genes and Genomes (KEGG) public database is utilized for diverse studies of genes, metabolites, and enzymes for the construction of large, comprehensive, and complex network diagrams. In this study, differential metabolites were classified in terms of the signaling pathways in the relevant species, *Ovis aries*. The comprehensive KEGG pathway diagram for the LTF group, compared with the CTF group, is shown in Appendix A. In the bubble plot, the rich factor refers to the ratio of the number of differential metabolites to the total number of metabolites identified in the corresponding pathway. Larger values represent a greater degree of enrichment, and the pathway with the most significant enrichment is identified as the signaling pathway with a *p*-value closest to zero. The dimensions of each bubble represent the number of metabolites enriched in the corresponding pathway. Metabolic pathways related to lipid metabolism have more enriched metabolites, and their rich factors were relatively high. These pathways include glycerophospholipid metabolism and linoleic acid metabolism.

### 3.5. Volatile Flavor Compounds in Tail Fat

As shown in Table 2, the aldehyde level was high among volatile compounds in sheep muscle fat, and aldehyde substances are derived mainly from unsaturated fatty acids, which create oxidative products (e.g., oleic acid and linoleic acid) after cooking that directly affect meat quality [15]. In this study, the nonanal and decanal contents were increased significantly in the LTF group relative to the CTF group (*p* < 0.05). The contents of the alcohols 1-hexanol and 1-octanol were significantly higher in the LTF group than in the CTF group (*p* < 0.05). Among acids and esters, the contents of octanoic acid, nonanoic acid, and ethyl ester were much higher in the LTF group than in the CTF group (*p* < 0.05). Additionally, the LTF group showed a significantly higher limonene content in hydrocarbon analysis (*p* < 0.05).

## 4. Discussion

*Lactiplantibacillus plantarum* is widely used as a probiotic in animal husbandry. Previous studies have demonstrated that *Lactiplantibacillus plantarum* can significantly improve blood indicators in both in vitro simulated gastrointestinal tract experiments and in vivo animal experiments, and a variety of benefits for animals have been identified. Blood is a substance that flows through the whole body, and changes in the concentrations of substances related to lipid metabolism in blood will affect the lipid metabolism of sheep. In the present study, the concentration of TG was reduced in the LTF group, which was consistent with previous research [16]. In addition, *Lactiplantibacillus plantarum* treatment reduced the plasma TG level by approximately 10% [17,18]. Lowering of TG levels with probiotic treatment may be attributable to the production of lipase, which breaks large fat molecules into simple and easily digestible substrates. The LTF group showed a notable reduction in TC, predominantly driven by the observed decrease in the HDL concentration [19]. The HDL concentration was decreased in the LTF group, and the LDL concentration was greatly decreased at the same time. Therefore, the ratio of HDL to LDL was much greater in the LTF group than in the CTF group. Previous research has shown that the LDL concentration is negatively correlated with health [20], and *Lactiplantibacillus plantarum* supplementation reduced LDL concentrations in Sunit sheep. The decrease in LDL concentration and increase in HDL-to-LDL ratio suggest that *Lactiplantibacillus plantarum* may help reduce the disease-associated state of metabolic syndrome. HDL can transport excess cholesterol from peripheral tissues or cells to the liver for catabolism via blood circulation [21]. The mechanism underlying this effect of *Lactiplantibacillus plantarum* may be reducing the absorption of bile acids in the enterohepatic circulation and increasing the catabolism of cholesterol into bile acids to reduce TC levels, rather than inhibiting the rate of cholesterol synthesis [22]. Meanwhile, a lower HDL-to-TC ratio indicates a reduced level of HDL available to transfer cholesterol from peripheral tissues, possibly allowing more cholesterol to be deposited in tail fat.

Gene expression is related to food intake. Delta 5 and delta 6 desaturases are encoded by the FADS1 and FADS2 genes, respectively, and have been confirmed to play roles in the biosynthesis of polyunsaturated fatty acids (PUFAs) [1]. In the present study, the expression of FADS1 was significantly reduced in the LTF group relative to the CTF group, which may be related to the low content of γ-linolenic acid in the tail fat of LTF sheep [23]. In addition, FADS1 encodes a delta 5 desaturase that catalyzes the conversion of C20:4n-3 to C20:5n-3, which may be a driver of the high EPA content in tail fat. FASN is involved in the synthesis of fatty acids in sheep, and its expression is positively correlated with the contents of PUFAs, especially EPA, in muscle [1]. SCD can catalyze the conversion of saturated fatty acids into monounsaturated fatty acids, and higher SCD activity is associated with higher monounsaturated fatty acid content in muscles [15]. All analyzed metabolites and genes with differential expression between the two groups are shown in KEGG map oas01100 (Appendix A). In that figure, upregulated expression in the LTF group relative to the CTF group is shown in red, and downregulated expression is shown in blue. As shown in the KEGG map oas01040 (Appendix A), EPA, gamma-linolenic acid, SCD, and FADS2 are involved in the biosynthesis of unsaturated fatty acids. All of these genes and substances may be involved in the “Fatty acid elongation in endoplasmic reticulum” (M00415) pathway. SCD and FADS2 can affect the synthesis of EPA and gamma-linolenic acid by altering the expression of upstream genes and enzymes.

Functional substances in the diet can impact human health and metabolism, and sheep crude tail fat contains some such metabolites. Specific functional fat metabolites were elevated in the LTF group. As the main components of cell membranes and organelle membranes, PUFAs participate in and regulate numerous biological activities in eukaryotic cells. EPA is a multifunctional omega-3 fatty acid that has cardiovascular benefits, including lowering plasma TG and non-HDL cholesterol levels. In particular, EPA may reduce the risk of atherosclerosis [24]. Tail fat also contains 16-HPA (omega-HPA), a fatty acid with cytostatic and cell death-inducing effects in the human lung adenosquamous carcinoma cell line H596 and adenocarcinoma cell line A549 [25]. Hepatic PC (16:0/16:0) provides a major source of phospholipids in steatohepatitis, affects organelle membrane integrity, and prevents fatty liver from promoting fat metabolism [26]. Cytidine 5’-diphosphocholine, an intermediate in the synthesis of PC, reduced aspartate aminotransferase, alanine aminotransferase, and lactate levels in blood samples collected from re-perfused rats. Cytidine 5’-diphosphocholine can also protect the liver from damage, preserve cellular mitochondrial function, and reduce oxidative stress [27]. β-nicotinamide D-ribonucleotide is a precursor of nicotinamide adenine dinucleotide, a central redox cofactor and substrate for key metabolic enzymes. Depletion of β-nicotinamide D-ribonucleotide is the causative factor of multiple human heredity diseases related to cellular metabolism in the human body [28]. Citrulline is not absorbed by the gut or liver and has the highly specific function of providing nitrogen to support protein homeostasis in peripheral tissues. Citrulline stimulates protein synthesis in skeletal muscle via the mammalian target of the rapamycin signaling pathway and regulates protein metabolism [29]. Studies of plasma and liver cholesterol levels indicate that chenodeoxycholate slightly increases the release of bile acid, affecting the body’s digestion and absorption of lipids. In conclusion, reducing the contents of chenodeoxycholate and DCA increases the contents of ketoisocaproic acid and succinate, with beneficial effects on the metabolism of glucose and lipids in humans. Phosphorylcholine is an essential nutrient required by microorganisms to modulate host immunity [30]. Cytosine methylation and demethylation play significant roles in regulating genomic DNA expression. Epigenetic modifications of cytosine and its successive oxidation products have potential effects on gene regulation and transcription [31]. Gamma-linolenic acid is a polyunsaturated fatty acid that reduces lipid deposition, and it decreases lipid content, increases beta-oxidation of fatty acids, decreases TC and cholesterol contents, and increases mRNA and protein expression of carnitine palmitoyltransferase 1A (CPT1A) and PPARA [32]. The traditional method of consuming sheep crude tail fat is to extract the oil, and loss of functional substances during this process has rarely been reported to date.

During the heat treatment of lamb meat, fatty acids, amino acids, phospholipids, and other alkaline substances are thermally decomposed and converted into various volatile flavor substances after oxidation. This process produces simple compounds such as alcohols, aldehydes, ketones, acids, and esters, which ultimately improve the sensory qualities of the meat, as well as heterocyclic compounds containing oxygen, sulfur, and nitrogen (such as thiophene and its derivatives). Volatile flavor compounds in the tail fat of the LTF group showed unique advantages over the CTF group. Nonanal not only has a fatty aroma but also improves the nutritional value of edible oil [33]. Dry-aged beef tenderloin has a unique nonanal flavor and high levels of lipid and protein oxidation [34]. In particular, nonanal and decanal aldehydes are important natural components that enhance floral fragrance in foods. Marked changes in human electroencephalography activity occur with nasal exposure to nonanal and decanal aldehydes and olfactory stimulation [35]. In a lipid suspension extracted from erythrocyte membranes, 1-hexanol produced a fruity aroma that could be partitioned to either the bilayer surface or interior in a time-dependent manner [36,37]. 1-Octanol is obtained from the hydrogenolysis of coconut oil and palm kernel oil and has a lemon-like aroma. It is added to some foods as a food flavor and is also used as a raw material for the production of caprylaldehyde and caprylic acid [38,39]. Some studies have shown that octanol has a two-way regulatory effect on the membrane potential of rat pancreatic β-cells and enhances the activity of volume-regulated anion channels [40]. Caprylic acid is a monocarboxylate saturated fatty acid involved in ketogenic metabolism. Caprylic acid at non-toxic levels exerts neuroprotective and mitochondrial-protective effects, thereby reducing systemic toxicity in several neurodegenerative disease models [41]. Nonanoic acid-ethyl ester is an endogenous metabolite with a pleasant oily and brandy-like aroma [42]. The consumption of animal fats is second only to that of vegetable fats. Animal fats have unique flavors and medicinal properties that cannot be completely replicated by vegetable fats. In this study, feeding *Lactiplantibacillus plantarum* increased the contents of the volatile flavor substances responsible for excellent flavor in crude fat, which may help improve the acceptance of tail fat by the public.

## 5. Conclusions

Feed supplemented with *Lactiplantibacillus plantarum* can improve blood lipid metabolism in Sunit sheep. By regulating genes related to fat metabolism, this supplement influences the synthesis and decomposition of lipid metabolites such as fatty acids and, thus, the formation of volatile flavor compounds in tail fat. After supplementation with *Lactiplantibacillus plantarum*, the metabolites present in Sunit sheep crude tail fat provide key information about its quality.

## Figures and Tables

**Figure 1 foods-11-02644-f001:**
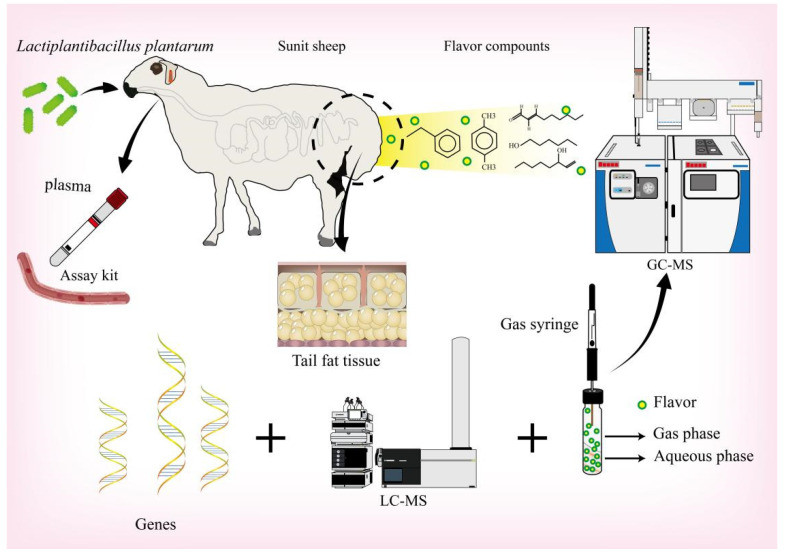
Experimental design diagram. Including animal experiments, RT-PCR experiments of genes related to lipid metabolism, GC-MS experiments of tail fat volatile flavor substances, and LC-MS metabolome experiments of tail fat.

**Figure 2 foods-11-02644-f002:**
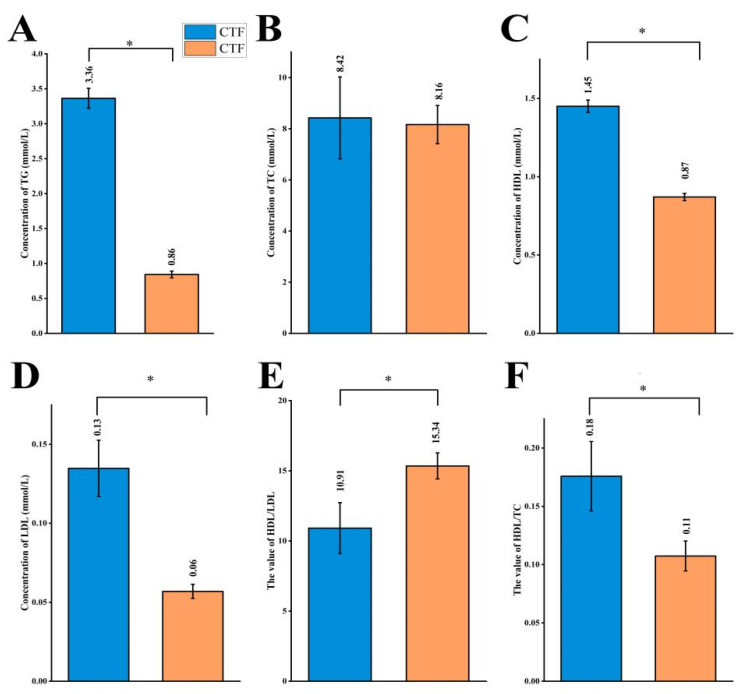
Plasma in CTF and LTF. (**A**) The concentration of TG of Sunit sheep in two feeding methods. (**B**) The concentration of TC of Sunit sheep in two feeding methods. (**C**) The concentration of HDL of Sunit sheep in two feeding methods. (**D**) The concentration of LDL of Sunit sheep in two feeding methods. (**E**) The value of HDL-to-LDL of Sunit sheep in two feeding methods. (**F**) The value of HDL-to-TC of Sunit sheep in two feeding methods. “*” indicates a significant difference between the two groups (*p* < 0.05).

**Figure 3 foods-11-02644-f003:**
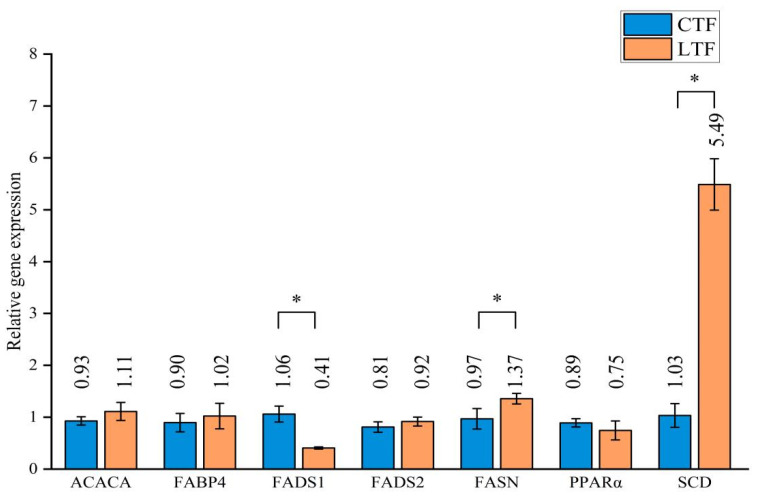
The relative expression of seven genes in the cecum. Data are expressed as average ± standard deviation. Note: CTF: control group (basal feed), LTF: experimental group (basal feed + Lactiplantibacillus plantarum). “*” indicates a significant difference between the two groups (*p* < 0.05).

**Figure 4 foods-11-02644-f004:**
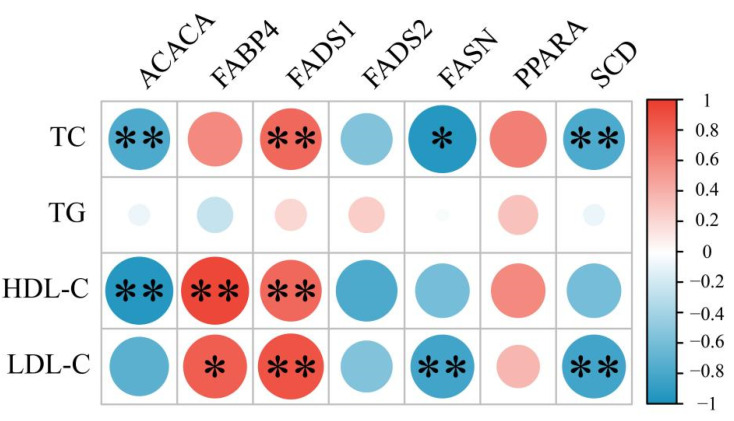
Heatmap of Spearman correlation coefficients between lipid parameters and lipid metabolism-related genes. “*” indicates a significant difference between the two groups (*p* < 0.05), and “**” indicates an extremely significant difference between the two groups (*p* < 0.01).

**Figure 5 foods-11-02644-f005:**
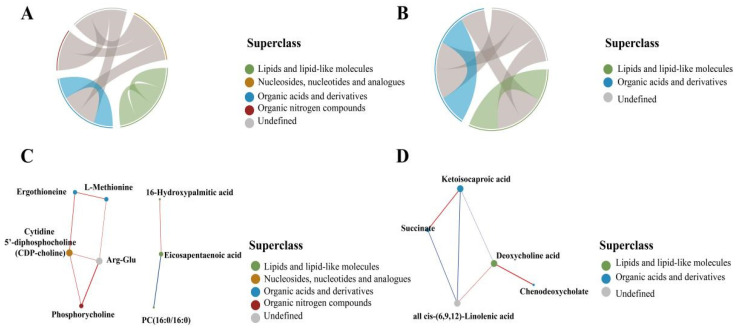
Chord diagram and network diagram for positive ion mode and negative ion mode. (**A**) Positive ion mode chord diagram. (**B**) Negative ion mode chord diagram. The starting point of the link in the inner circle in the Figure represents each significantly different metabolite, and the arc on the outer circle represents the classification of the significantly different metabolite. Colored lines indicate correlations within the various metabolites, and the lines are the same color as the subclasses. Dark grey lines indicate correlations between different classes of metabolites. (**C**) Positive ion mode network diagram. (**D**) Negative ion mode network diagram. The dots in the Figure represent significantly different metabolites. The size of the dots is related to the degree of connectivity. The greater the degree, the larger the dot. The colors of the lines represent correlations, red for positive correlations and blue for negative correlations. The thickness of the line represents the absolute value of the correlation coefficient. The thicker the line, the greater the correlation.

**Table 1 foods-11-02644-t001:** The different tail fat metabolic profiles between the LTF and CTF.

No	RT (min)	Metabolites	VIP	m/z	Adduct Ion	L/C Group	Pathways
F1	1.241	Undecanoic acid	2.954	185.154	(M−H)−	↓ *	Lipids and lipid-like molecules
F2	4.624	Deoxycholic acid	1.700	391.2842	(M−H)−	↓ *	
F3	1.776	Heptadecanoic acid	2.496	269.2478	(M−H)−	↓ *	
F4	4.349	Chenodeoxycholate	2.568	391.2839	(M−H)−	↓ *	
F5	1.588	Pelargonic acid	1.555	157.1229	(M−H)−	↑ *	
F6	1.243	Eicosapentaenoic acid	1.855	303.2317	(M+H)+	↑ **	
F7	1.142	16-Hydroxypalmitic acid	3.835	314.2689	(M+CH3CN+H)+	↑ **	
F8	5.177	Hepatic phosphatidylcholine (PC) (16:0/16:0)	2.661	756.553	(M+Na)+	↓ **	
F9	5.016	Stearoylcarnitine	3.836	428.373	(M+H)+	↓ *	
F10	4.973	(S)-(-)-Citronellic acid	2.842	212.164	(M+CH3CN+H)+	↑ *	
F11	7.779	Guanosine	2.038	282.084	(M−H)−	↑ *	Nucleosides, nucleotides, and analogues
F12	14.752	Cytidine 5’-diphosphocholine (CDP-choline)	2.408	489.1145	(M+H)+	↑ **	
F13	15.236	β-Nicotinamide D-ribonucleotide	1.441	335.0636	M+	↑ **	
F14	12.195	Succinate	1.527	117.019	(M−H)−	↑ *	Organic acids and derivatives
F15	1.190	Ketoisocaproic acid	3.890	129.055	(M−H)−	↑ *	
F16	12.863	L-Citrulline	1.620	176.1026	(M+H)+	↑ **	
F17	10.823	Ergothioneine	1.271	230.096	(M+H)+	↑ *	
F18	9.374	L-Methionine	1.097	150.058	(M+H)+	↑ *	
F19	15.423	Phosphorylcholine	7.547	184.073	(M+H)+	↑ **	Organic nitrogen compounds
F20	9.199	Choline	2.583	104.106	M+	↓ *	
F21	1.044	4-Pyridoxic acid	2.435	182.045	(M−H)−	↑ *	Organoheterocyclic compounds
F22	6.880	Cytosine	1.274	112.050	(M+H)+	↑ **	
F23	70.79	All cis-(6,9,12)-Linolenic acid	9.573	277.217	(M−H)−	↓ **	
F24	11.593	D-gluconate	1.517	195.051	(M−H)−	↑ *	Organic oxygen compounds

Note: RT(min) represents the retention time of the component. VIP represents the variable projection importance; the larger the value, the more important. m/z represents the mass-to-charge ratio. Adduct Ion represents the adduct ion information of the compound. Comparison of the number of corresponding flavor substances between the two groups. “↑” indicates that the amount in LTF is higher than in CTF, and “↓” indicates that LTF is lower than CTF. “*” indicates a significant difference between the two groups, and “**” indicates an extremely significant difference between the two groups.

**Table 2 foods-11-02644-t002:** Volatile compounds of Sunit sheep crude tail fat (μg/kg).

	Compound (μg/1 kg)	CTF	LTF	Significance
Aldehydes	Hexanal	25.23 ± 5.86	32.87 ± 14.97	NS
	Octanal	46.91 ± 8.95	68.71 ± 11.69	NS
	Nonanal	151.72 ± 12.74 ^b^	241.19 ± 23.98 ^a^	**
	Decanal	28.27 ± 2.55 ^b^	38.92 ± 3.70 ^a^	*
	2-Nonenal, (E)-	19.33 ± 0.89	21.90 ± 1.67	NS
Alcohols	1-Hexanol	19.11 ± 2.06 ^b^	32.03 ± 4.43 ^a^	*
	1-Octen-3-ol	32.87 ± 5.57	40.27 ± 0.37	NS
	2-Octen-1-ol,(E)-	22.92 ± 3.47	11.44 ± 4.79	NS
	1-Octanol	18.42 ± 1.51 ^b^	31.89 ± 2.91 ^a^	**
Acids	Octanoic acid	25.62 ± 5.55 ^b^	51.93 ± 9.83 ^a^	*
	Nonanoic acid	51.23 ± 17.48	98.15 ± 20.59	NS
	Undecanoic acid	32.41 ± 5.47	158.36 ± 64.89	NS
Esters	Allyl 2-ethyl butyrate	41.37 ± 2.99	39.10 ± 2.26	NS
	Acetic acid, nonyl ester	13.86 ± 0.78	15.50 ± 0.59	NS
	Nonanoic acid, ethyl ester	17.52 ± 1.12^b^	26.00 ± 2.13^a^	*
	Octanoic acid, ethyl ester	17.78 ± 4.86	21.88 ± 7.94	NS
	Hexanoic acid, ethyl ester	11.20 ± 1.55	14.30 ± 0.49	NS
Amines	Ethanolamine	36.67 ± 5.67	23.71 ± 6.63	NS
Aromatic hydrocarbons	Ethylbenzene	197.68 ± 8.78	178.35 ± 5.06	NS
Hydrocarbons	Styrene	21.79 ± 7.62	20.79 ± 4.43	NS
	Longifolene	14.98 ± 3.29	31.96 ± 6.73	NS
	Limonene	62.62 ± 5.46 ^b^	157.43 ± 18.55 ^a^	**

Note: Values are expressed as mean and standard deviation (n = 6). NS, nonsignificant. ^a,b^ Significant differences between feeding regimens (*p* < 0.05). * *p* < 0.05; ** *p* < 0.01.

## Data Availability

Data are contained within the article and Appendix A.

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
