# Peer review of "Impact of Feeding Probiotics on Blood Parameters, Tail Fat Metabolites, and Volatile Flavor Components of Sunit Sheep"

_foods, 2022, doi:10.3390/foods11172644_

Round 1

Reviewer 1 Report

Report on the manuscript foods- 1812447 entitled: Impact of feeding probiotics on blood indexes, tail fat metabolites and volatile flavor components of Snit sheep.

 Major comments:

-         -Table S2 is useless and is not mentioned once in the manuscript. Please, include its information within the text and remove the table.

-        -  Numeric values must be included in the Results section.
Figure 1 and lines 245-250. Please, include the different mean values and/or the % of the difference. Same for Figure 2 and lines 259-263.

-          -Lines 286-300 and Table S3. Please, delete; useless information.

-         - Why was not the library NIST used to identify more metabolites of the LC analysis? In table S4, a lot of unidentified compounds with VIP > 1 and P < 0.05 are shown.
Identification of compounds from LC analysis must be improved as well as Table 1 and Results and Discussion related parts.

-          -Regarding Figure S2, what is the difference between Fig. S2A and S2B? Why is not mentioned in the text?
In addition, since a clear separation among samples can be observed along to[1] in Figure 2A and 2B, a higher number of samples must be considered for analysis and several replicas must be also considered. The described results are related to the considered samples and cannot be considered as representative of a bigger population.

Other comments:

-         - Line 13. …compared to…

-         - Line 14-15. Was the metabolomics carried out using GC or LC?

-          -Line 26. …which live in …

-         - Line 29-30. Actually, the cited article says that the tail fat % varies between 5 and 15% depending on slaughter weight… Not 20%. Please, correct it.

-        -  Line 31. Please, remove “light”.

-       -   Lines 39-44. Please, remove this paragraph. Information regarding the history of Metabolomics and its basics is not necessary.

-         - Line 46. … to judge food quality.

-         - Lines 60-66. Please, delete. This comment does not belong to the Introduction. Perhaps it could belong to the Discussion.

-        -  Lines 66-71. Again, please, delete it. This comment does not belong to this section.

-          - Figure after line 74. This figure is a graphical abstract. Not sure whether the journal is or not interested in it. Please, remove it if necessary.

-        -  Line 106-107. Laboratory jugular vein? Meaning?

-         - Line 107. Blood samples were centrifuged at.... and plasma was stored at….

Reviewer 2 Report

In this paper, the effect of probiotics on fat metabolites and flavor of Sunit sheep tail fat was investigated by dietary supplementation of Lactobacillus Plantarum. Sunit sheep is a high-quality mutton sheep breed in Inner Mongolia, China, and its taste and flavor are very popular with consumers. However, there is a piece of adipose tissue in the tail of Sunit sheep, which has low food value and is generally used as a lubricant and surfactant. In this paper, the potential nutritional value of sheep tail fat was explored through the detection of metabolites and volatile flavor compounds in Sunit sheep. Overall, the manuscript is well written and the logical framework is well organized. This manuscript is recommended for publication in Foods with minor revisions.Specific comments/suggestions are given below:

1. Line 14:Please modify the format between "P" and the symbol, and modify it in          the full text.

2. Line 19:Please unify the expression of "L. Plantarum" in the full text

3. Line 39-43:Please provide references support.

4. Line 50-52:Please provide references support.

5. Line 74:Please change the picture of the full text into vector picture.

6. Line 90:Note the space between the number and the unit symbol.

7. Line 119:Please modify "rt-PCR" to "RT-PCR"

8. Line 201:For the first occurrence of the English abbreviation please change to the full name (Please check the full text).

9. Line 247:Please indicate the specific figure of the result.

10. Line 246-252:Please use the same expression for "Figure" (Please check the full text).

11. Line 271:Note the space between letters and symbols (Please check the full text).

Reviewer 3 Report

Respected Author,

Thank you very much for the possibility to read your interesting manuscript. After the ban of antibiotics and growth stimulators are probiotics and prebiotics on of the way how to preventively improve safety and income of food production and it is still interesting to examine the side effects of the probiotics feeding, for example on the meat quality etc.

I have some specific comments to your manuscript.

L9 – I recommend to specify fat as crude fat.

L14 – I recommend to explain abbreviations like FASN and SCD immediately when they are used in the manuscript. (in abstract for example here). Also this recommendation is for the whole manuscript.

L74 – the name and the label as figure 1 for the picture is missing.

L94-101 – there is missing information about the dry matter intake and frequency of feeding. Also for the better repeatability of your experiment I recommend to also mention the energy content of both diets.

L95 – I recommend to specify what type of corn was used as feed component – corn is too general (corn silage? corn grain? corn grain meal? High moisture corn?)

L265 – I recommend to explain CTF and LTF again in the abbreviations of Figure 2 – figures have to be self explainable.

L282 – This correlation was calculated in the whole data set? It is possible to split in according to the feeding group? Were there different correlation relationship between the lipid index and lipid metabolism-related genes in control and experimental group?

L324 – The table is not self explainable. Add in to the table footnote abbreviations for RT, m/z, L/C. What does arrow up and down mean? What does the star mean? It is statistically significance? Which test was used (it could be stated in the footnote).

L364 – Figure 4 – I do not know if I have some issue with the figure, however it is unreadable. There are very small letters.

In general, the manuscript is prepared very well, however some figures are unreadable. I really like the proportion of the current literature resources used for the introduction and discussion from the last 5 years.

I wish you all the best in your future experiments.

Best Regards,

Reviewer

Reviewer 4 Report

Additional remarks:

Unfortunately, the scientific quality of this manuscript was unsatisfactory. I found lot of problematic parts in this manuscript.

Detailed review:

Introduction:

lines 27-28: need rewriting this sentence! Good flavour, but what has a good flavour? Meat or fat? What does genetic performance stability mean? What does other characteristics mean?

lines 39-44: please focus on relevant literatures!

line 49 and other places: “Lactobacilus Plantarum”: correct name: Lactobacilus plantarum !

lines 57-60: rather this fit to the materials and methods section!

line 66: “The study found…” which study? Please add reference!

lines 74-75: this figure is not necessary in this section. Is this a graphical abstract?

Materials and methods

line 82: what does “disease-free” animals? May be these a special group?

line 88: CFT or CTF? Nevertheless, why indicate CTF and LTF, easier if use: CG and LG!

line 90: why add the L. plantarum at night?

line 94: how and when offer the basal diet (concentrate)? Was it ad libitum?

line 103: “adult” not adult, 3-month-old the beginner age, plus 90-day experiment, so all animals’ age is 6 months!

lines 104-105: “Human slaughter” what does it mean? Slaughter need meet the current rules and regulations! Where fed the animals? So, evening was slaughtered the animals? Please clarify it!

lines 105-107: this sentence is very incomprehensible! Moreover, “plasma samples were centrifuged” I think the blood samples were centrifuged!

line 114: why used “lipid index” when analysed lipid parameters and their ratios!

lines 115-116: why indicate HDL-C and LDL-C? Why not HDL and LDL, these are the accepted abbreviations of these parameters.

Results:

all Figures are not visible!!

lines 249-250: “decreased faster”; “decreased slower” these are incorrect word connection, because not investigated the tendency! Two groups were examined, at the end of the investigation period, where samples were taken!

Figure 1 title: not serum rather plasma (see Materials and methods section)!

Figure 2: please add the full name of genes in the footnote!

Figure 4: unfortunately, this figure is unconvincing and not interpretable!

Discussion:

lines 413-417: serum or plasma?

lines 416-417: what kind of marker? “It is composed of free form and HDL-C”? This is not clear!!!

line 424: what does “plays a probiotic role” mean?

lines 506-508: suggest delete this sentence, not to important part to this section.

lines 509-510: “mutton fat”, “mutton tail fat” ? better simple tail fat!

Conclusion:

Very short this section, please improve this section based on own results!

Round 2

Reviewer 1 Report

The authors have carried out all the reviewers' comments

Reviewer 4 Report

The introduction, materials and methods, results, discussion and conclusions sections  of this manuscript were extensively improved by authors, so I recommend this manuscript for publishing in the Foods journal.
